# Simultaneous Effects of Carboxyl Group-Containing Hyperbranched Polymers on Glass Fiber-Reinforced Polyamide 6/Hollow Glass Microsphere Syntactic Foams

**DOI:** 10.3390/polym14091915

**Published:** 2022-05-07

**Authors:** Jincheol Kim, Jaewon Lee, Sosan Hwang, Kyungjun Park, Sanghyun Hong, Seojin Lee, Sang Eun Shim, Yingjie Qian

**Affiliations:** 1Department of Chemistry and Chemical Engineering, Education and Research Center for Smart Energy and Materials, Inha University, Michuhol-gu, Incheon 22212, Korea; 22192178@inha.edu (J.K.); 22192186@inha.edu (J.L.); bulls@inha.edu (S.H.); 22201542@inha.edu (K.P.); 2LG Electronics Inc. 51, Gasan Digital 1-ro, Geumcheon-gu, Seoul 08592, Korea; sanghyun.hong@lge.com (S.H.); seojin710.lee@lge.com (S.L.)

**Keywords:** syntactic foams, hyperbranched polymer, polyamide 6, hollow glass microsphere, lubricant, compatibilizer, composites

## Abstract

The hollow glass microsphere (HGM) containing polymer materials, which are named as syntactic foams, have been applied as lightweight materials in various fields. In this study, carboxyl group-containing hyperbranched polymer (HBP) was added to a glass fiber (GF)-reinforced syntactic foam (RSF) composite for the simultaneous enhancement of mechanical and rheological properties. HBP was mixed in various concentrations (0.5–2.0 phr) with RSF, which contains 23 wt% of HGM and 5 wt% of GF, and the rheological, thermal, and mechanical properties were characterized systematically. As a result of the lubricating effect of the HBP molecule, which comes from its dendritic architecture, the viscosity, storage modulus, loss modulus, and the shear stress of the composite decreased as the HBP content increased. At the same time, because of the hydrogen bonding among the polymer, filler, and HBP, the compatibility between filler and the polymer matrix was enhanced. As a result, by adding a small amount (0.5–2.0 phr) of HBP to the RSF composite, the tensile strength and flexural modulus were increased by 24.3 and 9.7%, respectively, and the specific gravity of the composite was decreased from 0.948 to 0.917. With these simultaneous effects on the polymer composite, HBP could be potentially utilized further in the field of lightweight materials.

## 1. Introduction

The international automobile market has been changing from traditional internal combustion engines to eco-friendly vehicle development, in line with corporate average fuel efficiency (CAFÉ) standards and automobile greenhouse gas (GHG) emission standards [1]. As economic and the environmental concerns for fuel consumption have evolved, lightweight materials became of great interest to the automotive industry [2,3,4,5,6,7]. In order to apply lightweight materials in the automotive industry, the mechanical strength and the processability of the material must be accompanied by reduced specific gravity.

To reduce the specific gravity of the polymeric composite materials, numerous studies have been actively conducted on hollow glass microsphere (HGM)-containing lightweight syntactic foams (SFs) in various applications, such as automotive, marine, and aerospace [8,9,10,11,12,13,14,15]. HGM exhibits a low specific gravity and high electrical and thermal insulation properties. However, the addition of HGMs in higher contents results in the weakening of mechanical properties of SFs [12,16]. Addition of fibrous material, such as glass fiber (GF) or carbon fiber to SFs, can enhance the mechanical properties of the SFs; these are named reinforced syntactic foam (RSF) [17,18,19]. Nevertheless, the addition of rigid HGM and GF particles have a detrimental effect on the processability by increasing viscosity. Furthermore, due to the high shear of the extrusion process, HGM particles are broken, and this breakage results in an increase in the specific gravity of the polymer composites [9,20].

To address these issues, lubricating agents and plasticizers have been widely studied [21,22,23,24]. These materials modify the viscosity of the polymer melt, and reduce the friction; as a result, they increase the processability of the polymer composite with their addition in small amounts to the composites. The paraffin waxes, esters, and fatty acids derivatives are commonly used as lubricants for polymers [25]. Generally, when these typical lubricants are added to the composite material, the mechanical strength of the material weakens, since these lubricants have low molecular weights and low compatibility with polymer matrices [26,27]. Recently, topology-engineered polymers, such as star-shaped polymers and hyperbranched polymers have been studied, in order to be applied as lubricants [24,28].

Hyperbranched polymer (HBP) is a highly branched three-dimensional polymer. The HBP molecule has a low degree of entanglement and low viscosity due to the dendritic architecture of the molecule [29,30]. It can be utilized as a rheological modifier in polymeric composites by acting as molecular ball-bearings at the expense of the van der Waals forces [31]. According to WanG′s study [24], as the polymer chain topology changes from a linear to a hyperbranched structure, the intrinsic viscosity and the complex viscosity of the polymer decrease significantly, and the shear stability of the polymer increases as well. Besides, the abundant functional groups in the HBP molecules make it possible to form hydrogen bonding among HBP, polymer matrix, and the fillers in the composite material [32,33]. This functionality results in the compatibilization of the filler in the composite material, and could make up for the weakened mechanical strength of the syntactic foams. Gu et al. [34] increased the toughness and the mechanical strength of soybean protein film with addition of hyperbranched polyester, which forms strong hydrogen bonding with the soybean protein matrix. Peng et al. [35] improved the wettability and the interfacial adhesion of carbon fiber to an epoxy resin matrix by poly(amido amine) functionalization.

As a result of its dendritic structure and abundant functional groups, HBP could act as a lubricating additive and as a compatibilizer at the same time in the GF-reinforced SF system. In the present study, the carboxyl group-containing HBP molecule was introduced to RSF composite material to reduce the viscosity and to improve the mechanical properties of the composite material simultaneously. We used polyamide 6 (PA 6) as a polymer matrix, since PA 6 is widely used in the automotive industry, and HBP can be dispersed effectively in the PA 6 matrix due to the hydrogen bonding ability of PA 6. Scanning electron microscopy (SEM) microphotographs revealed that the compatibility between filler and PA 6 polymer matrix was enhanced with the addition of a small amount of HBP (0.5–2.0 phr). The density of the RSF and ashes of the RSF were measured to calculate HGM breakage of RSF material. Tensile strength increased significantly from 59 to 73 MPa, and the specific gravity decreased from 0.948 to 0.917 with the addition of HBP.

## 2. Materials and Methods

### 2.1. Materials

Injection grade Nylon-6 (Ultramid, B3S, Ludwigshafen, Germany), with a density of 1.13 g/cm^3^ and melt volume rate of 160 cm^3^/min at a temperature of 275 °C, given load of 5 kg, was supplied by BASF (Ludwigshafen, Germany). HGM (IM16K) provided by 3M (Saint Paul, SP, USA), with a true density of 0.46 g/cm^3^, a crush strength of 110 MPa, and average diameter of 20 μm, was used as the lightweight filler in this study. The chopped nylon-compatible-sized GF used for a reinforced filler was 995-10P grade from Owens Corning (Toledo, OH, USA). The GF grade has a nominal diameter of 10 μm, chopped length of 4 mm, and sized with amino silane coupling agent. Carboxyl group functionalized HBP, CYD-7010 (synthesized from adipic acid and hyperbranched polyester with 98:2 molar ratio), with a melt range of 135~155 °C, was supplied by Weihai CY Dendrimer Technology Corporation (Weihai, China). The chemical structures of PA 6, HGM, GF, and HBP are presented in Figure 1.

### 2.2. Methods

#### 2.2.1. Sample Preparation

The syntactic foams were prepared in a co-rotating twin-screw extruder (TEK-25, SM PLATEC Co., Ansan, Korea), with screw diameter of 25 mm, D_o_/D_i_ = 1.55, L/D = 40, comprising two kneading zones for both dispersive and distributive mixing. Twin screw configuration is depicted in Appendix A. In order to minimize hydrolysis during the extrusion process, Nylon-6 was pre-dried at 80 °C for 24 h. The PA 6 pellets and HBP powder were fed into the main hopper first. Then, the HGM was fed into the first side feeder, after which the GF was fed into the second side feeder. All composites were melt-mixed at the same conditions under 200 rpm at a temperature range of 240/240/250/250/250/250/250/250/250/250 °C from hopper to die. The PA 6 and the HBP were pre-melted before introducing HGM fillers. The extrudates were passed through a hot water bath, pelletized, and dried at 80 °C for 24 h before injection molding. Specimens for the tensile, flexural testing, and density measurements were injection molded in a lab-scale injection molding machine (VDCII-50, JINHWA GLOTECH, Cheonan, Korea) with a clamping force of 50 tons. Injection molding was carried out at a barrel temperature profile of 190/220/230/240/240 °C from hopper to nozzle, and a mold temperature of 80 °C. The compositions and the codes of the prepared samples are reported in Table 1.

#### 2.2.2. Characterization

Complex viscosity and shear modulus were measured using an Anton Paar MCR 302 rheometer (Graz, Austria) with 25-mm parallel plate geometry. Extrudates of neat Nylon-6 and composites were molded at 240 °C. Small angle oscillatory shear (SAOS) frequency sweep tests were performed at a constant temperature of 240 °C within a linear viscoelastic regime in the range of 0.1–100 rad/s.

The tensile-fractured samples for morphological observation were characterized with field-emission scanning electron microscopy (FE-SEM) using the model S-4300 from HITACHI (Tokyo, Japan). The fractured surface was sputter-coated with platinum. Right after the tensile testing, the fractured surface of each sample was dipped in liquid nitrogen for 10 min to keep the shape of the surface.

The differential scanning calorimetry (DSC) analysis was conducted with NETZSCH DSC200F3 (Selb, Germany), with a sample mass of 3 mg that was set in an aluminum pan with a cover under an N_2_ atmosphere with a flow rate of 40 mL/min, to determine the crystallization behavior of the composites. The glass transition temperature (*T_g_*), melting temperature (*T_m_*), and melt crystallization temperature (*T_c_*) were measured from the second heating and cooling cycle, respectively. The profile of the thermogram is depicted in Appendix A. The samples were first heated from 20 to 250 °C at a rate of 30 °C/min and held at 250 °C for 4 min to eliminate thermal history of the samples. The samples were then cooled down to −10 °C at a rate of 30 °C/min and held at −10 °C for 10 min. The DSC data were collected in the second cycle (segments 6 and 8 as heating and cooling, respectively). The samples were heated at a rate of 10 °C/min from −10 to 250 °C, and then held at 250 °C for 4 min. Then the samples were cooled to −10 °C at a rate of 10 °C/min. The samples were finally held at −10 °C for 4 min.

The crystallinity of the samples was calculated by the following equation:(1)$$X_{c}\;\left(\%\right)=\frac{∆H_{f}}{W_{m}\cdot ∆H_{o}}$$
where $∆H_{f}$ is heat of fusion for the sample, $W_{m}$ is the mass fraction of the PA 6, and the $∆H_{o}$ is the heat of fusion of the theoretical 100% crystalline PA 6 (240 J/g).

The non-isothermal crystallization behavior of the composites was calculated using the Avrami equation [36,37,38]:(2)$$1-V_{t}=\exp\left(-Z_{t}t^{n}\right)$$
where *V*_*t*_ is the relative volumetric fraction of crystalline, *Z*_*t*_ is the overall crystallization rate constant which reflects both nucleation and crystal growth, and *n* is the Avrami index. The *V_t_* can be obtained by the following equation:(3)$$V_{t}=\frac{W_{c}}{W_{c}+\left(\frac{\rho_{c}}{\rho_{a}}\right)\left(1-W_{c}\right)}$$
where *W_c_* = ∆*H*(*t*)/∆*H_total_* is the crystalline mass fraction, and *ρ_c_* and *ρ_a_* are crystalline density (1.20 g/cm^3^) and amorphous density (1.09 g/cm^3^) of the PA 6, respectively [39]. When the crystallization rate constant is corrected by taking the cooling rate into account for the Avrami equation, which was applied only to the isothermal crystallization, the following equation is used, where the *Z_c_* is the crystallization rate constant in the non-isothermal condition:(4)$$logZ_{c}=\frac{logZ_{t}}{dT/dt}$$

Thermogravimetric analysis (TGA) was performed with PerkinElmer TGA4000 (Waltham, MA, USA) under an N_2_ atmosphere with a flow rate of 20 mL/min. The samples with a mass of 10 mg were contained in an aluminum pan, and measured at a heating rate of 10 °C/min from 40 to 800 °C to determine thermal stability of the composite samples.

The tensile strength (ASTM D638) and flexural modulus (ASTM D790) of each specimen were performed using a universal testing machine (DUT-2TC, DAEKYUNG ENGINEERING Co., Bucheon, Korea) with a 2-ton load cell. Specimen dimensions for the tensile test are presented in Appendix A. Samples for the flexural test having dimensions of 127 × 12.7 × 6.4 mm^3^ (length × width × thickness) were employed for the test in accordance with ASTM D790. The tensile tests were performed at a cross-head speed of 50 mm/min. The flexural tests were carried out in the three-point bending configuration at a cross-head speed of 1.54 mm/min. Flexural modulus was calculated from the slope of the initial linear portion of the curves. The five different samples were tested for accurate results. Two types of density measurements were conducted with a hydro-densimeter (GP300S, MATSUHAKU, Taichung, Taiwan) and a gas pycnometer (BELPycno, MicrotracBEL Corp., Osaka, Japan) to measure the bulk density of the composites and residue inorganic ashes of the composites, respectively.

## 3. Results and Discussion

### 3.1. Rheological Properties

The rapid increase in complex viscosity in RSF control is shown in Figure 2a due to the high content of fillers (HGM, GF). As the contents of HBP increased, the complex viscosity of the composites was reduced across the entire frequency range. It was confirmed that the complex viscosity of RSF-COOH 2.0 decreased by 4.9 times at 0.1 rad/s compared to the neat RSF due to the addition of the HBP molecules in the composite system. Figure 2b shows that the shear stress decreased as the HBP content increased. The internal friction was alleviated by HBP, which has a low intrinsic viscosity in the molten state. HBP molecules improved the polymer chain mobility due to its lubricating effect, and this improved polymer chain mobility in filled RSF composite is on account of the dendritic architecture of HBP [40]. The reduced shear stress could contribute to the reduction in HGM breakage arising from the high shear in the extrusion process and decrease in the specific gravity of the composite.

Changes in storage modulus (G′) and loss modulus (G″) of the composites for different contents of HBP in RSF are shown in Figure 2c,d. In overall frequency, the G″ was higher than the G′ value, which means the composites were measured in the predominantly viscous region. Kang et al. [41] found that the syntactic foams aggregated at higher content (higher than HGM 10 wt%), and the viscoelastic response changed from viscous to elastic behavior at low frequency. Similarly, both the storage modulus (G′) and the loss modulus (G″) of the RSF were increased compared to the neat PA 6. When HBP was added to the RSF, the G′ and G″ decreased in overall frequency. Due to the low entanglement degree of the HBP molecule itself, the lubricating effect was applied in the composite system; as a result, the overall G′ and G″ tend to decrease in RSF-COOH composites. At high frequencies, the alteration of segmental dynamics was not observed because the entanglement structure of the polymer was retained [42]. As a result, the effect of the lubrication which comes from the dendritic architecture of the HBP molecule was predominant. However, at low frequencies the slope of the G′ to the frequency was decreased in all samples compared to the RSF, and the G′ value slightly increased when 0.5 phr of HBP was added. These phenomena come from the abundant functional groups in the HBP molecule. When the entanglement of the polymer chain was released at low frequency, the unravelling of the entanglement was hindered by the hydrogen bonding among the PA 6, fillers, and the HBP molecules. A similar result was observed in Bhardwaj’s research [43]. This could be evidence that not only hydrogen bonding, but also lubrication, were functioning in the RSF-COOH composite system.

### 3.2. Morphology

SEM microphotographs of the fractured surfaces of the composite materials after tensile testing are presented in Figure 3. In Figure 3a, without HBP, most of the HGMs are exposed to the fractured surface. However, when the HBP was added to the RSF (Figure 3b–d), the partially exposed and almost buried HGMs were observed, indicating that HBP increased the interfacial adhesion between polymer matrix and HGM. The enlarged microphotographs of the GFs are presented in Figure 3e–h. The GF surface of the neat RSF was smooth without attached PA 6 matrix. On the other hand, when HBP was added, the GF surface was covered with PA 6. On account of the enhanced interfacial adhesion of the polymer matrix to the fillers, which comes from the hydrogen bonding among the polymer matrix, fillers, and HBP, the compatibility of the fillers in the PA 6 matrix was increased, and the mechanical strength of the RSF-COOH composites could be enhanced compared to the neat RSF.

It would be useful to discuss the fracture mechanism of the composite material to investigate the change of the mechanical properties of the composites with and without HBP addition. When the external load was applied to the composite, micro-cracks were generated near the filler surface due to weak interfacial adhesion with the polymer matrix [18]. Without HBP, when a high content of HGM was added, the sites of cracks increased; furthermore, the cracks propagated easily because there were neighboring cracks from other HGM particles. Since the strength of the matrix is larger than that of the interface between the HGM and polymer matrix [44], cracks on the surface of the HGM propagated easily. On the other hand, when HBP was added to the RSF composite, the filler-polymer adhesion increased due to the increased hydrogen bonding sites that came from the abundant functional groups of the HBP molecule. As a result, under the external load, the effect of the cracks on the surface of the HGMs decreased and the effect of the plastic deformation and yielding became predominant. Therefore, the mechanical strength of the RSF-COOH was higher than that of the neat RSF composite.

### 3.3. Crystallization Behavior

The cooling curves and the DSC data of the neat PA 6, PA 6/HGM, and PA 6/GF composites at temperatures ranging from 205 to 180 °C are shown in Appendix A. The crystallization temperature of the PA 6/HGM 20 composite decreased compared to the neat PA 6. However, the change in crystallization temperature of the PA 6/HGM 5 and the PA 6/GF 5 samples was not significant. However, high content of HGM particles, which was associated with our RSF system, influenced the crystallization behavior significantly. The crystallization rate constant Z_t_ decreased, and the t_1/2_ of the PA 6/HGM 20 increased. These phenomena reflect that the crystallization of the RSF without HBP was delayed since the HGM particles in the PA 6 matrix could act as the steric obstacle for the crystalline development of the PA 6 polymer [45]. The *T_g_* data of the PA 6, RSF, and RSF-COOH samples are presented in Appendix A. The *T_g_* of the samples decreased from 60.81 to 45.52 °C as the HGM and GF were introduced to the PA 6; this was due to the weak interfacial adhesion between the fillers and PA 6 [46]. However, when HBP was added to the RSF system, the *T_g_* of the composites slightly increased from 45.52 to 46.70 °C. This indicates that HBP addition enhanced the interfacial adhesion of the fillers with the PA 6 matrix.

Figure 4a presents the DSC thermograms of the neat RSF and RSF-COOH composites, and the DSC data are listed in Table 2.

As can be seen in Table 2, the crystallization temperature gradually increased as the concentration of HBP increased. However, the crystallinity *X_c_* increased as the filler content increased to 1.0 phr, and then decreased when the content of HBP further increased. A similar trend was observed in ZhanG′s study [47]. HBP can form hydrogen bonding with the PA 6 matrix and the filler surface, and the HBP molecules could act as the nucleating agent on the filler surface. When a low content of HBP was added to the composite (0–1.0 phr), crystallization of the PA 6 was accelerated by the increased amount of nuclei. As the HBP content was further increased beyond 1 phr, the growth of the crystal may be hindered by the excessive HBP molecules; as a result, the crystallinity was slightly reduced in RSF-COOH 2.0 compared to the RSF-COOH 1.0. The crystallization behavior of the RSF-COOH composites would affect the mechanical properties of the composites.

The crystallization kinetics of syntactic foams in Appendix A showed that the crystallization rate was slowed by the further addition of HGM. The degree of crystallinity of the RSF composites calculated by the Avrami equation is depicted in Figure 4b. As demonstrated in Table 3, t_1/2_ of the RSF-COOH composite decreased due to a higher crystallization rate constant as the content of HBP increased. HBP can be responsible for the crystallization rate, which was accelerated by not only enhanced chain mobility, but also by increased nucleation sites.

### 3.4. Thermogravimetric Analysis

The extruded pellets were analyzed by TGA, which is presented in Figure 5. The residual char amount after the temperature reached 800 °C was 30 wt% in all composites, indicating the inorganic filler content in the RSF composites. The onset temperature of the thermal decomposition where the weight percent was 95% was increased from 408 to 415 °C when the HBP content increased from 0 to 0.75 phr. This result indicates that the thermal stability of the RSF composites was improved due to the enhanced interfacial adhesion between the filler and polymer matrix through the compatibilization which resulted from HBP [48]. When the HBP content was further increased to 2.0 phr, the decomposition occurred earlier at 403 °C. As a result of the lower thermal stability of HBP itself, which is presented in Appendix A, the excessive amount of HBP affects the thermal stability of the RSF-COOH composites where the content of HBP is higher than 1.0 phr.

### 3.5. Mechanical Properties and Specific Gravity of the Composite

Appendix A show that increasing weight fraction of HGMs from 0 to 20 wt% did affect the mechanical properties of the composite significantly. When the weight fraction of HGM reached to 20 wt%, the tensile strength was significantly decreased by 40.6% due to the poor dispersion of the filler and the weak interfacial adhesion of the filler to the polymer matrix (Appendix A).

Figure 6 and Table 4 show the mechanical properties and the specific gravity of the PA 6, RSF control, and HBP containing RSF. The tensile strength increased by 24.3% in RSF-COOH 0.75 compared to the neat RSF. As a result of the abundant functional groups that can form hydrogen bonding, the strong interfacial adhesion between PA 6 polymers and fillers could enhance the tensile strength of the RSF-COOH composite. On the other hand, as the HBP content increased to 2.0 phr, tensile strength gradually decreased. It is interpreted that the physical property was weakened by the lubricating effect that comes from the dendritic structure of the HBP molecules. We can conclude that the optimal HBP content where the tensile strength is a maximum is 0.75–1.0 phr, and this indicates that the compatibility and the lubrication would be complementary at this content.

The flexural test results of the composite are summarized in Figure 6b and Table 4. HBP had a positive effect on the flexural modulus and flexural strength. In RSF-COOH 1.0, compared to RSF-control, the flexural modulus increased from 4032 to 4424 MPa, and the flexural strength increased from 50 to 55 MPa. The increase in flexural strength is a result of the high interfacial adhesion of the HBP molecules. Furthermore, on account of the reduced filler breakage during the processing, which is summarized in Table 5, foam-core sandwich structure of the RSF-COOH composites could be formed more effectively than that with the neat RSF.

Specific gravity data of the RSF samples are provided in Figure 6c. It was shown that the specific gravity continued to decrease as the content of HBP increased. This is because the hollow HGMs can be damaged not only by the high shear in the extrusion process, but also by the additional injection molding. HBP brings out economic feasibility through tribological advantage, resulting in a much lighter RSF composite due to reduced HGM breakage. HGM breakage was calculated by TGA and gas pycnometer analysis in Table 5 and Appendix A. HGM breakage of the neat RSF and RSF-COOH 2.0 was 9.14 and 2.84 vol%, respectively, which show good agreement with the tendency of the specific gravity in Figure 6c. These results can be supplementary evidence that the remarkable variation in shear stress, mentioned in Figure 3c, improved the internal friction of the composite.

## 4. Conclusions

In this study, glass fiber-reinforced PA6 syntactic foam material was prepared with the addition of the carboxyl group-containing HBP for lightweight material applications. The morphological, rheological, and thermal properties of the composite material were characterized. It was revealed that HBP could increase mechanical properties, and could act as a rheological modifier simultaneously. The outcomes from the present study are summarized as follows:(1)In the rheological characterization, the complex viscosity and the shear stress of the RSF-COOH composites decreased across the entire frequency range compared to the neat RSF composite without HBP. The storage modulus and loss modulus tended to decrease with increasing content of HBP in the composite; however, the slope of the storage modulus at low frequency tended to decrease. These results are evidence that the lubricating effect due to the dendritic structure of HBP, in addition to hydrogen bonding, were functioning in the RSF-COOH composite at the same time.(2)The tensile strength and flexural modulus were increased by 24.3 and 9.7%, respectively, in RSF-COOH composite compared to the neat RSF. In the FE-SEM microphotographs, it was shown that the compatibility of the filler improved, and that filler-polymer adhesion enhanced when HBP was added. In addition, in DSC data, the crystallinity of the composite (*X_c_*) increased to 22.54% with the addition of HBP in the composite. The enhanced compatibility of the filler with the polymer matrix, and crystallization behavior contributed to the increased mechanical strength of the RSF-COOH composite material.(3)As a result of the reduced shear stress of the RSF-COOH composite in the high shear-applied extrusion process, HGM breakage considerably decreased from 9.14 to 2.84%. As a result, the specific gravity of the composite significantly decreased from 0.948 to 0.917 when HBP was introduced to the RSF composite.

The results show that HBP plays multiple roles in the RSF composite: modifying rheological properties, increasing mechanical properties, and reducing specific gravity. With these simultaneous effects in the RSF composite, HBP could be applied further in the lightweight materials field.

## Figures and Tables

**Figure 1 polymers-14-01915-f001:**
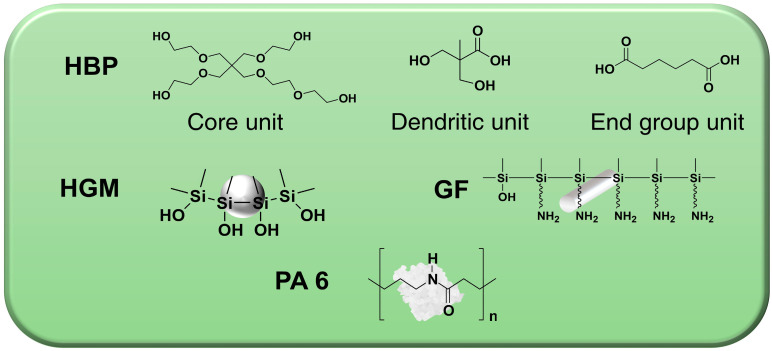
Chemical structures of PA 6, HGM, GF, and HBP.

**Figure 2 polymers-14-01915-f002:**
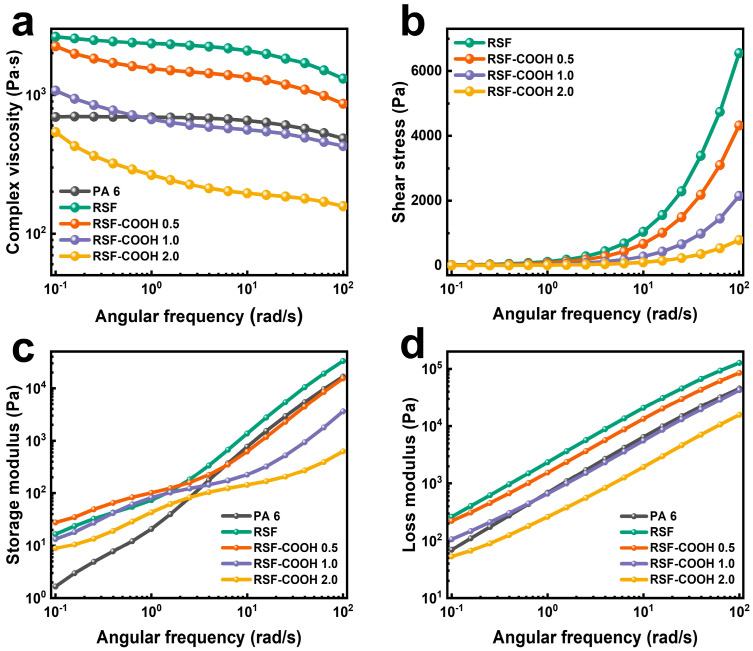
Complex viscosity (**a**), shear stress (**b**), storage modulus (**c**), and loss modulus (**d**) of the neat PA6 and RSF composites with different HBP contents.

**Figure 3 polymers-14-01915-f003:**
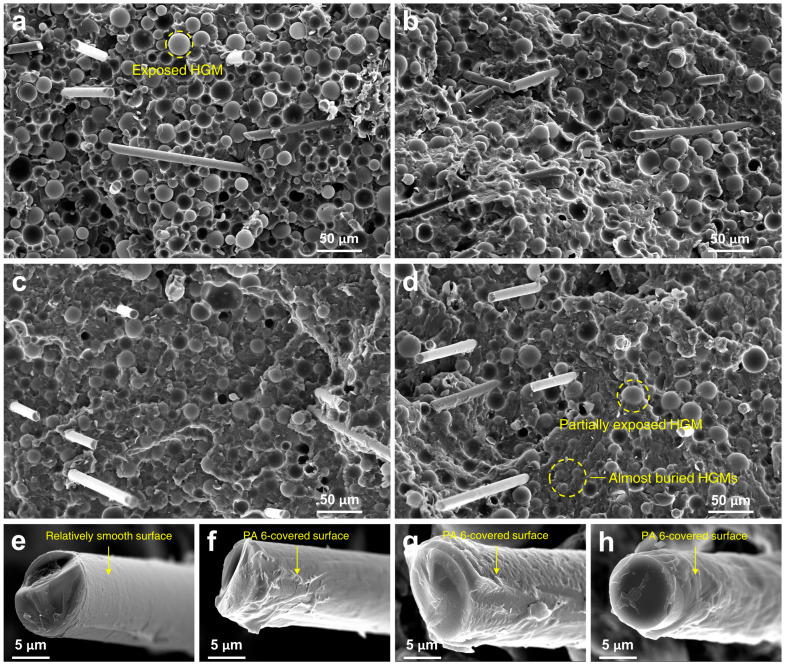
SEM microphotographs of fractured surface of the composite samples after tensile testing: (**a**) RSF control, (**b**) RSF-COOH 0.5, (**c**) RSF-COOH 1.0, and (**d**) RSF-COOH 2.0. Enlarged microphotographs of the GFs which are in the (**e**) RSF control, (**f**) RSF-COOH 0.5, (**g**) RSF-COOH 1.0, and (**h**) RSF-COOH 2.0, respectively.

**Figure 4 polymers-14-01915-f004:**
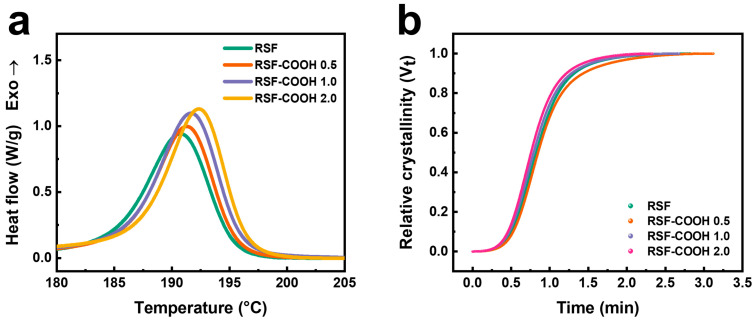
DSC thermograms cooled at 10 °C/min (**a**) and plots of relative crystallinity (*V_t_*) vs time (**b**) for non-isothermal crystallization of RSF composites with various HBP contents.

**Figure 5 polymers-14-01915-f005:**
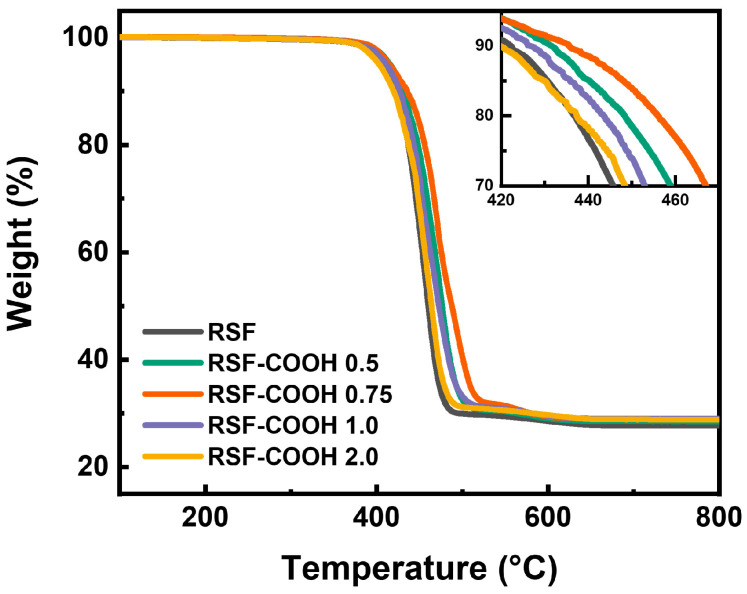
Thermogravimetric analysis data for RSF containing various HBP contents. Inset: A close-up of the 420–470 °C region.

**Figure 6 polymers-14-01915-f006:**
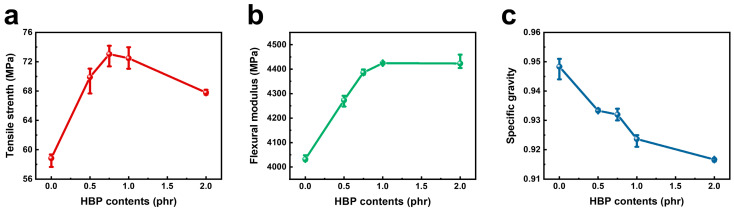
Tensile strength (**a**), flexural modulus (**b**), and the specific gravity (**c**) of the RSF with various contents of HBP.

**Table 1 polymers-14-01915-t001:** Composition and identification details of prepared samples referred to as syntactic foams (SF), consisting of different PA 6, HGM, GF, and HBP contents.

Sample Label	PA 6 (wt%)	HGM (wt%)	GF (wt%)	HBP (phr ^a^)
RSF	72	23	5	-
RSF-COOH 0.5	71.6	23	5	0.5
RSF-COOH 0.75	71.5	23	5	0.75
RSF-COOH 1.0	71	23	5	1.0
RSF-COOH 2.0	70.6	23	5	2.0

^a^ Parts per hundred of PA 6 resin.

**Table 2 polymers-14-01915-t002:** DSC data for RSF containing various HBP contents under non-isothermal crystallization.

Sample Label	$T_{m}$ (°C)	$∆H_{f}$ (J/g)	$X_{c}$ (%)	$T_{c}$ (°C)	$T_{onset}$ (°C)	$∆H_{c}$ (J/g)	$∆H_{c}^{*}$ (J/g)
RSF	221.3	34.68	20.00	190.8	195.1	40.98	56.70
RSF-COOH 0.5	220.8	35.00	20.40	191.3	195.4	41.16	57.57
RSF-COOH 1.0	221.1	38.39	22.54	191.7	195.7	43.30	61.01
RSF-COOH 2.0	221.0	37.34	21.86	192.4	196.2	42.36	59.51

**Table 3 polymers-14-01915-t003:** Crystallization kinetic parameters of non-isothermal crystallization data for RSF containing various HBP contents.

Sample Label	n	$Z_{t}$	$Z_{c}$	t_1/2_ (min)	Adj. R-Square
RSF	3.8	1.376	1.032	0.84	0.9997
RSF-COOH 0.5	4.1	1.337	1.029	0.85	0.9997
RSF-COOH 1.0	3.8	1.633	1.050	0.80	0.9997
RSF-COOH 2.0	3.9	1.972	1.070	0.76	0.9997

**Table 4 polymers-14-01915-t004:** Comparison of the mechanical properties of neat PA 6 and RSF containing various HBP contents.

Sample Code	Specific Gravity	Tensile Strength (MPa)	Elongation (%)	Flexural Modulus (MPa)	Flexural Strength (MPa)
PA 6	1.125	80.2 ± 2.6	13.6 ± 13.8	2843 ± 73	35.3 ± 0.9
RSF	0.948	58.8 ± 1.2	1.4 ± 0.2	4032 ± 16	50.0 ± 0.2
RSF-COOH 0.5	0.933	69.9 ± 2.2	1.4 ± 0.6	4273 ± 26	53.0 ± 0.3
RSF-COOH 0.75	0.932	73.1 ± 1.7	1.3 ± 0.3	4386 ± 13	54.6 ± 0.2
RSF-COOH 1.0	0.924	72.5 ± 1.5	1.8 ± 0.6	4424 ± 0	54.9 ± 0
RSF-COOH 2.0	0.917	70.7 ± 1.5	1.3 ± 0.2	4423 ± 36	54.6 ± 0.4

**Table 5 polymers-14-01915-t005:** HGM breakage measurement results of each RSF composite containing various HBP contents.

Sample Code	Ash Density ^a^ (g/cm^3^)	Residual HGM Density (g/cm^3^)	HGM Breakage (vol%)
RSF	0.5829	0.4772	9.14
RSF-COOH 0.5	0.5590	0.4596	6.07
RSF-COOH 1.0	0.5463	0.4504	3.73
RSF-COOH 2.0	0.5431	0.4470	2.84

^a^ Measured by exposing the pellets to 550 °C for 2 h to remove the matrix (PA 6). The true density of the residual inorganic ashes consisting of HGMs and GFs was measured by gas pycnometer.

## Data Availability

Not applicable.

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
