# Peer review of "Simultaneous Effects of Carboxyl Group-Containing Hyperbranched Polymers on Glass Fiber-Reinforced Polyamide 6/Hollow Glass Microsphere Syntactic Foams"

_polymers, 2022, doi:10.3390/polym14091915_

Round 1

Reviewer 1 Report

The carelessness of the present manuscript makes this reviewer wonder who is indeed under evaluation.

The manuscript needs of severe correction on several sentences (TGA does not determine thermal properties but thermal stability; tests are not repeated five times but five different specimens were tested, and so on…); and/or systematic wrong terms (processibility), or the non-use of SI units.

Some issues under the Experimental Section are correctly described unlike other ones are poorly, or just omitted. Nothing is said about the HBP properties (at 250 ºC under inert atmosphere, it seems to lose up to a 50% in weight), or how it is incorporated into the composites; tensile or flexural specimen dimensions are never specified, neither temperature tests; why are cryogenic the tensile fracture surfaces observed in SEM …? Setting plots in Figure 2 is wrong with respect to both the foot and the references to this figure under the Results and Discussion section; DSC runs should need to be well described and records in support included; and so and so on.

About the Results and Discussion Section, definitely it is not acceptable for this referee. Sometimes because it is poor, as for example SEM micrographs discussion (by the way, it only refers to a sample); sometimes because they are just general comments, as for example "rheological properties" discussion. And some others because they are just wrong, as those comments dealing with the HGM content in the composites, which no way is a manuscript's concern, or the absence of any mention to the glass transition of the PA6 and if varies or not in the different compositions.

Author Response

Reviewer 1

The carelessness of the present manuscript makes this reviewer wonder who is indeed under evaluation.

The manuscript needs of severe correction on several sentences (TGA does not determine thermal properties but thermal stability; tests are not repeated five times but five different specimens were tested, and so on…); and/or systematic wrong terms (processibility), or the non-use of SI units.

Response:

The revised parts are as follows:

Line: 161

“The samples with a mass of 10 mg were measured at a heating rate of 10 °C/min from 40 to 800 °C to determine thermal stability of the composite samples”.

Line: 172

“The five different samples were tested for accurate results”.

Lines: 37, 47, and 53

“processability”.

Line: 94

“crush strength of 110 MPa”.

Some issues under the Experimental Section are correctly described unlike other ones are poorly, or just omitted. Nothing is said about the HBP properties (at 250 ºC under inert atmosphere, it seems to lose up to a 50% in weight)

Response:

According to the TGA data of HBP (Figure S8), the residual weight was 73.8% at 250 oC under inert atmosphere (26.2% weight loss). When the HBP was introduced to the RSF composite system, the environment changed compared with the TGA system. In addition, as presented in Figure 5, at 250 °C, there was no significant weight loss indicating that in composite system, the HBP degradation was not observed.

how it is incorporated into the composites; tensile or flexural specimen dimensions are never specified, neither temperature tests;

Response:

Lines: 110–114

“The PA 6 pellets and HBP powder were fed into the main hopper firstly. Then, the HGM was fed into the first side feeder, then the GF was fed into the second side feeder. All composites were melt-mixed at the same condition under 200 rpm at a temperature range of 240/240/250/250/250/250/250/250/250/250 °C from hopper to die. The PA 6 and the HBP were pre-melted before introducing HGM fillers”.

Lines: 166–169

“Specimen dimension for the tensile test is presented in Figure S1 (b). Samples for the flexural test having a dimension of 127  12.7  6.4 mm3 (length  width  thickness) were employed for the test in accordance with ASTM D790.’ The specimen dimension for the tensile test has been added to the Figure S1 (b)”.

why are cryogenic the tensile fracture surfaces observed in SEM …?

Response:

Lines: 131–135

“The tensile-fractured samples for the morphological observation were characterized with field emission scanning electron microscopy (FE-SEM) using the model S-4300 from HITACHI (Japan). The fractured surface was sputter-coated with platinum. Right after the tensile testing, the fractured surface of each sample was dipped in liquid nitrogen for 10 min to keep the shape of the surface”.

Setting plots in Figure 2 is wrong with respect to both the foot and the references to this figure under the Results and Discussion section;

Response:

The arrangement of the Figure 3 has been changed:

DSC runs should need to be well described and records in support included; and so and so on.

Response:

Line: 136–139

“The differential scanning calorimetry (DSC) analysis was conducted with NETZSCH DSC200F3 (Germany), with a sample mass of 3 mg, which was set in aluminum pan with a cover, under the N2 atmosphere with a flow rate of 40 ml/min, to determine the crystallization behavior of the composites.”

Lines: 142–148

“The samples were first heated from 20 °C to 250 °C at a rate of 30 °C/min and held at 250 °C for 4 min to eliminate thermal history of the samples. The samples were then cooled down to -10 °C at a rate of 30 °C/min and held at -10 °C for 10 min. The DSC data was collected in second cycle (segment 6 and 8 as heating and cooling, respectively). The samples were heated at a rate of 10 °C/min from -10 °C to 250 °C, and then held at 250 °C for 4 min. Then the samples were cooled to -10 °C at a rate of 10 °C/min. The samples were finally held at -10 °C for 4 min.”

Lines: 161–163

“The samples with a mass of 10 mg were contained in alumina pan, and measured at a heating rate of 10 °C/min from 40 to 800 °C to determine thermal stability of the composite samples”.

About the Results and Discussion Section, definitely it is not acceptable for this referee. Sometimes because it is poor, as for example SEM micrographs discussion (by the way, it only refers to a sample); sometimes because they are just general comments, as for example "rheological properties" discussion.

Response:

Lines: 217–228

 “SEM microphotographs of the fractured surfaces of the composite materials after tensile testing are presented in Figure 3. In Figure 3 (a), without HBP, most of the HGMs are exposed to the fractured surface. However, when the HBP was added to the RSF (Figure 3 (b–d)), the partially exposed and almost buried HGMs were observed, indicating that the HBP increased the interfacial adhesion between polymer matrix and HGM. The enlarged microphotographs of the GFs were presented in Figure 3 (e–h). The GF surface of the neat RSF was smooth without attached PA 6 matrix. On the other hand, when HBP was added, the GF surface was covered with the PA 6. Because of the enhanced interfacial adhesion of the polymer matrix to the fillers, which comes from the hydrogen bonding among the polymer matrix, fillers, and HBP, the compatibility of the fillers in the PA 6 matrix was increased, and the mechanical strength of the RSF-COOH composites could be enhanced compared to the neat RSF.”

The previous sentences or paragraphs have been deleted or modified:

Previous Line180: “The reduction of viscosity could greatly contribute to the improvement of the dispersibility of HGM and GF in the melt blending with high shear rate.” – deleted

Current Line181–184: “It was confirmed that the complex viscosity of RSF-COOH 2.0 decreased by 4.9 times at 0.1 rad/s compared to the neat RSF due to the addition of the HBP molecules in the composite system.”

And some others because they are just wrong, as those comments dealing with the HGM content in the composites, which no way is a manuscript's concern, or the absence of any mention to the glass transition of the PA6 and if varies or not in the different compositions.

Response:

The Tg data of the PA 6, RSF, RSF-COOH 0.5, RSF-COOH 1.0, and RSF-COOH 2.0 have been added in Figure S6.

Lines: 260–266

“The Tg data of the PA 6, RSF, and RSF-COOH samples are presented in Figure S6. The Tg of the samples was decreased from 60.81 °C to 45.52 °C as the HGM and GF were introduced to the PA 6 due to the weak interfacial adhesion between the fillers and PA 6 [46]. However, when the HBP was added to the RSF system, the Tg of the composites were slightly increased from 45.52 °C to 46.70 °C. This indicates that HBP addition enhanced the interfacial adhesion of the fillers with PA 6 matrix”.

Reviewer 2 Report

The reviewer believes that the manuscript can be published in this journal, but some improvements are needed.

General comments, the manuscript need to be better discussed and it is important compare and correlate each analysis performed between the different analysis. 

  1. Include the Tg of different conditions samples using DSC and TGA methods;
  2. Include the geometry or standard of the samples used in tensile and flexural tests;
  3. It is important to include SEM images of all conditions samples (add
    RSF-COOH 1.0 and RSF-COOH 2.0).

Author Response

Reviewer 2

The reviewer believes that the manuscript can be published in this journal, but some improvements are needed.

General comments, the manuscript need to be better discussed and it is important compare and correlate each analysis performed between the different analysis. 

  1. Include the Tg of different conditions samples using DSC and TGA methods.

Response:

The Tg data of the PA 6, RSF, RSF-COOH 0.5, RSF-COOH 1.0, and RSF-COOH 2.0 have been added in Figure S6.

Lines: 260–266

“The Tg data of the PA 6, RSF, and RSF-COOH samples are presented in Figure S6. The Tg of the samples was decreased from 60.81 °C to 45.52 °C as the HGM and GF were introduced to the PA 6 due to the weak interfacial adhesion between the fillers and PA 6 [46]. However, when the HBP was added to the RSF system, the Tg of the composites were slightly increased from 45.52 °C to 46.70 °C. This indicates that HBP addition enhanced the interfacial adhesion of the fillers with PA 6 matrix”.

  1. Include the geometry or standard of the samples used in tensile and flexural tests;

Response:

Lines: 166–169
“Specimen dimension for the tensile test is presented in Figure S1 (b). Samples for the flexural test having a dimension of 127  12.7  6.4 mm3 (length  width  thickness) were employed for the test in accordance with ASTM D790.”

  1. It is important to include SEM images of all conditions samples (add RSF-COOH 1.0 and RSF-COOH 2.0).

Response:

The SEM images of all samples have been added to Figure 3.

Lines: 217–228

 “SEM microphotographs of the fractured surfaces of the composite materials after tensile testing are presented in Figure 3. In Figure 3 (a), without HBP, most of the HGMs are exposed to the fractured surface. However, when the HBP was added to the RSF (Figure 3 (b–d)), the partially exposed and almost buried HGMs were observed, indicating that the HBP increased the interfacial adhesion between polymer matrix and HGM. The enlarged microphotographs of the GFs were presented in Figure 3 (e–h). The GF surface of the neat RSF was smooth without attached PA 6 matrix. On the other hand, when HBP was added, the GF surface was covered with the PA 6. Because of the enhanced interfacial adhesion of the polymer matrix to the fillers, which comes from the hydrogen bonding among the polymer matrix, fillers, and HBP, the compatibility of the fillers in the PA 6 matrix was increased, and the mechanical strength of the RSF-COOH composites could be enhanced compared to the neat RSF.”

Reviewer 3 Report

The manuscript be Kim et al. describes the addition of a carboxylate-containing hyperbranched polymer (HBP) to a syntactic composite based on hollow glass microspheres (HGM) in Nylon-6, reinforced with glass fibres.  It is shown that the HBP improves the flow properties, resulting in reduced breakage of of the HGMs, lower density and improved mechanical properties of the composites.

In my opinion, the manuscript is generally well written, the data is presented clearly and supports the conclusions drawn.  I found only one minor technical issue:

Regarding Fig. 3: Please indicate how the surfaces were prepared (e.g. tensile or flexural testing).

There is clearly a difference between the surfaces in Fig. 3a and 3b.  The hollow glass microspheres (HGMs) appear more exposed in the former, compared with the latter.  However, it is not easy to see any difference in the distributions of HGMs between those two images.  Would it be possible to add a description of what features to look for in the images, perhaps with some indications (arrows pointing to, or circles enclosing the features) on the images themselves, please?

Also, it would be helpful to expand the caption to highlight the differences between the specimens in Fig. 3c to 3f, please.

There were also some minor technical issues or typographical errors:

It looks like some of the elements in Fig. 1 have been distorted during manuscript preparation.

L128: The phrase 'The cryogenic samples following tensile-fractured for the morphological observation...' is unclear and difficult to understand.

L177: Should be 'Changes in storage modulus....and loss modulus...'

L216: 'Worthful' usually applies to people, rather than things or concepts.  It would be better to say 'important' or 'useful'.  Also, 'about' is not required in that sentence.  I suggest: 'It would be useful to discuss the fracture mechanism...'  

L283-284:  'was' is not required: '...compatibilization which resulted from HBP [47].'

L313:  Something seems to be missing from this phrase.  I suggest it should be: '...sandwich structure could be formed more effectively than with the neat RSF.'

L166 and L344: I suggest it would be better to use an expression such as '...across the entire frequency range'.

I recommend 'minor corrections' to address these points.

Author Response

Reviewer 3

The manuscript be Kim et al. describes the addition of a carboxylate-containing hyperbranched polymer (HBP) to a syntactic composite based on hollow glass microspheres (HGM) in Nylon-6, reinforced with glass fibres. It is shown that the HBP improves the flow properties, resulting in reduced breakage of of the HGMs, lower density and improved mechanical properties of the composites.

In my opinion, the manuscript is generally well written, the data is presented clearly and supports the conclusions drawn.  I found only one minor technical issue:

Regarding Fig. 3: Please indicate how the surfaces were prepared (e.g. tensile or flexural testing).

Response:

Lines: 217–228

“SEM microphotographs of the fractured surfaces of the composite materials after tensile testing are presented in Figure 3…”

There is clearly a difference between the surfaces in Fig. 3a and 3b. The hollow glass microspheres (HGMs) appear more exposed in the former, compared with the latter. However, it is not easy to see any difference in the distributions of HGMs between those two images. Would it be possible to add a description of what features to look for in the images, perhaps with some indications (arrows pointing to, or circles enclosing the features) on the images themselves, please?

Response:

Figure 3 has been modified with yellow dashed-circles, arrows, and description of the pointed features.

Also, it would be helpful to expand the caption to highlight the differences between the specimens in Fig. 3c to 3f, please.

Response:

Figure 3. SEM microphotographs of fractured surface of the composite samples after tensile testing: (a) RSF control, (b) RSF-COOH 0.5, (c) RSF-COOH 1.0, and (d) RSF-COOH 2.0. The enlarged microphotographs of the GFs: (e) RSF control, (f) RSF-COOH 0.5, (g) RSF-COOH 1.0, and (h) RSF-COOH 2.0, respectively. The PA 6 was attached on the GF surface in (f-h), and GF in (e) had a smooth surface without PA 6.

There were also some minor technical issues or typographical errors:

It looks like some of the elements in Fig. 1 have been distorted during manuscript preparation.

Response:

Figure 1 has been modified.

L128: The phrase 'The cryogenic samples following tensile-fractured for the morphological observation...' is unclear and difficult to understand.

Response:

Lines: 131–135

“The tensile-fractured samples for the morphological observation were characterized with field emission scanning electron microscopy (FE-SEM) using the model S-4300 from HITACHI (Japan). The fractured surface was sputter-coated with platinum. Right after the tensile testing, the fractured surface of each sample was dipped in liquid nitrogen for 10 min to keep the shape of the surface”.

L177: Should be 'Changes in storage modulus....and loss modulus...'

Response:

Line: 191

“Changes in storage modulus (G’) and loss modulus (G”) …”

L216: 'Worthful' usually applies to people, rather than things or concepts.  It would be better to say 'important' or 'useful'.  Also, 'about' is not required in that sentence.  I suggest: 'It would be useful to discuss the fracture mechanism...'  

Response:

Lines: 234

“It would be useful to discuss the fracture mechanism of the composite…”

L283-284:  'was' is not required: '...compatibilization which resulted from HBP [47].'

Response:

Lines: 305–308

“…compatibilization which resulted from HBP [48].”

L313:  Something seems to be missing from this phrase.  I suggest it should be: '...sandwich structure could be formed more effectively than with the neat RSF.'

Response:

Lines: 336–339

“…sandwich structure of the RSF-COOH composites could be formed more effectively than that of the neat RSF”.

L166 and L344: I suggest it would be better to use an expression such as '...across the entire frequency range'.

Response:

Lines: 180–181

“As the contents of HBP increased, the complex viscosity of the composites was reduced across the entire frequency range”.

Lines: 369–371

“In the rheological characterization, the complex viscosity and the shear stress of the RSF-COOH composites were decreased across the entire frequency range compared to the neat RSF composite without HBP”.

Round 2

Reviewer 1 Report

In the opinion of this referee, the serious flaws detected at the first version of this manuscript remain unsolved at the revised version, as long as they should need something more than just a mere make-up job to be overcome.

In consequence the manuscript must be rejected.